# Application of Bacteriophages for Mycobacterial Infections, from Diagnosis to Treatment

**DOI:** 10.3390/microorganisms9112366

**Published:** 2021-11-16

**Authors:** Christopher G. Shield, Benjamin M. C. Swift, Timothy D. McHugh, Rebekah M. Dedrick, Graham F. Hatfull, Giovanni Satta

**Affiliations:** 1Department of Pathobiology and Population Sciences, Royal Veterinary College, Hatfield AL9 7TA, UK; bswift@rvc.ac.uk; 2Centre for Clinical Microbiology, University College London, London NW3 2PF, UK; t.mchugh@ucl.ac.uk (T.D.M.); g.satta@ucl.ac.uk (G.S.); 3Biological Sciences, University of Pittsburgh, Pittsburgh, PA 15260, USA; dedrick@pitt.edu (R.M.D.); gfh@pitt.edu (G.F.H.)

**Keywords:** mycobacteriophage, phage, mycobacterium, tuberculosis, TB, BCG, MAP, diagnostics, therapy, proof-of-concept

## Abstract

Mycobacterium tuberculosis and other non-tuberculous mycobacteria are responsible for a variety of different infections affecting millions of patients worldwide. Their diagnosis is often problematic and delayed until late in the course of disease, requiring a high index of suspicion and the combined efforts of clinical and laboratory colleagues. Molecular methods, such as PCR platforms, are available, but expensive, and with limited sensitivity in the case of paucibacillary disease. Treatment of mycobacterial infections is also challenging, typically requiring months of multiple and combined antibiotics, with associated side effects and toxicities. The presence of innate and acquired drug resistance further complicates the picture, with dramatic cases without effective treatment options. Bacteriophages (viruses that infect bacteria) have been used for decades in Eastern Europe for the treatment of common bacterial infections, but there is limited clinical experience of their use in mycobacterial infections. More recently, bacteriophages’ clinical utility has been re-visited and their use has been successfully demonstrated both as diagnostic and treatment options. This review will focus specifically on how mycobacteriophages have been used recently in the diagnosis and treatment of different mycobacterial infections, as potential emerging technologies, and as an alternative treatment option.

## 1. Introduction

Mycobacterial infections are responsible for some of the most deadly and difficult to control infections in humans and animals. Tuberculosis (TB), caused primarily by *M. tuberculosis*, is thought to infect over 10 million people each year and causes the death of at least 1.6 million people annually worldwide [1]. The highest burden of these cases is attributed to low–middle-income countries (LMICs). Opportunistic mycobacterial infections in people are also caused by a range of non-tuberculous mycobacteria (NTM), including members of the *M. avium* complex (MAC) and *M. abscessus* complex (MAB). Other more recently identified and rarer NTM diseases include *M. chimaera* infections in cardiothoracic patients following exposure to contaminated heater-cooler units, unusual NTMs infections in immunocompromised hosts and BCGosis, a rare disseminated granulomatous disease, following intravesical *Bacillus Calmette-Guérin* (BCG) immunotherapy and in patients with predisposing genetic conditions such as the Mendelian susceptibility to mycobacterial diseases (MSMD) [2,3]. In veterinary medicine, *M. bovis* is the primary cause of TB in cattle and other animals and causes 140,000 new cases and 11,400 deaths per year globally [1].

Bacteriophages are viruses that infect bacteria and are the most abundant lifeforms on earth [4]. There are two types of phages with distinct lifecycles, lytic phages and temperate phages. Lytic phages infect, replicate and break open their host, whereas temperate phages can enter the lytic lifecycle or establish lysogeny by stably maintaining their DNA in the host either by integration into the host chromosome or as an extracellular replicon, and repressing lytic gene expression [5]. Bacteriophages can have very narrow host ranges, infecting specific subspecies of bacteria, but can also have relatively broad host ranges, capable of infecting several bacteria genera. Bacteriophages’ ability to kill their host make them attractive tools to treat infections and, although there is a body of literature on their clinical use in Eastern Europe, the Western world has only started to discover their potential benefits [6].

This review will focus on two separate but complementary topics: the use of bacteriophages for the diagnosis of mycobacterial infections and their use as alternative treatment options for challenging cases.

## 2. Mycobacteriophages as Diagnostics

TB control is limited by current diagnostics. Clinicians are still reliant on X-rays, microscopy and cultures as universal tools to diagnose TB [7]. Molecular platforms, such as the GeneXpert system, have made a difference to diagnosing TB by shortening the time to detection and improving sensitivity [8]; however, they are not yet considered a universal tool for diagnosis [9], due to the associated cost per test/scale up, need for well-trained/ consistent staff and need of a stable power source [10]. Culturing mycobacteria is generally seen as the gold standard diagnostic; however, many mycobacterial pathogens are slow-growing, for example, *M. tuberculosis* and *M. bovis* can take up to 12 weeks to culture on solid media, and *M. avium* subspecies *paratuberculosis* (MAP) can take up to 16 weeks. Culture is also insensitive, because relatively high numbers of bacilli are required for visible growth. The slow growth and low sensitivity makes the use of solid culture as a diagnostic for TB infections both impractical and inefficient [11]. The introduction of automated liquid culture systems, endorsed by the WHO, has improved the practicality and accessibility of culture as a primary diagnostic, but it still remains slow and relatively expensive [12].

Many of the current diagnostics for TB infections are immunologically based, where the host response is used to diagnose infection. A major problem with this approach is that mycobacteria are generally characterized by their ability to avoid their host’s immune system, which can result in the inconsistent detection of infected individuals [13], particularly where the pathogen effectively evades immunity. Succinctly, the methods may fail to detect infection due to the pathogen’s innate evasion of host immune responses.

Molecular methods such as PCR exist to detect mycobacterial pathogens to overcome the reliance on immune response. However, widely used PCR platforms (such as GeneXpert) are expensive and tend not to have the required sensitivity to detect *M. tuberculosis* in a range of matrices due to the inefficient lysis of mycobacteria as well as potential inhibitors that are often found in samples being tested. The development and deployment of rapid, sensitive diagnostics is a cornerstone of strategies to understand, control, and eradicate TB [1]. New diagnostics for mycobacterial infections need to be appropriate for use in LMICs or in agricultural settings—meaning low-cost, simple and robust. Therefore, by developing diagnostics that advance the speed, sensitivity, simplicity and cost of testing, TB control can be strengthened.

Phage-based diagnostics historically consisted of two general areas: phage amplified biologically (PhAB) assay and phage reporter assays (PRAs). PhAB exploits a certain aspect of the phage’s natural ability to infect, amplify and break open cells to detect the mycobacteria; PRAs typically involve genetically modified bacteriophages or their hosts so that a fluorescent, luminescent or alternative signal can be detected. A meta-analysis of the PhAB assay found that its main limitation was a high rate of indeterminate/contaminated results (20%) [14]. However, this technique’s appropriateness for LMICs has long been recognized [14,15]. PRAs have consistently had high sensitivity and specificity, although the only effort to commercialize the technology—The Bronx Box (Sequella, Rockville, USA)—has been discontinued [16]. Recent PRA endeavors have focused on facilitating the technology by creating affordable detection equipment [17]. PhAB assays and PRAs have been apprised in detail elsewhere [18,19,20]. Therefore, we sought to evaluate other, less explored, phage-based approaches. For more in-depth detail on PhAB and PRA methodologies, readers are encouraged to see the FASTPlaque TB™ (Biotech Labs Ltd., Ipswich, UK) [16,21] and proof-of-concept luciferase reporter phage assays [20,22]. Table 1 presents a summary of commercial and published phage technologies that have been used to detect mycobacteria.

PhAB assays have also been demonstrated for other mycobacteria, including *M. ulcerans*, *M. avium*, *M. scrofulaceum, M. marinum, M. fortuitum* and *M. chelonae* [23], although these have not been exploited further.

The advantages of phage-based approaches are reflected by improvements achieved in the speed, specificity and sensitivity. All assays give results faster than the “gold standard” eight weeks and consistently high specificity values are reported. However, directly comparing sensitivities is difficult because different comparators have been used. Some studies used culture, whereas others used GeneXpert (Cepheid, Sunnyvale, CA, USA), and fewer still used sputum smear microscopy. Standardizing reference tests would improve confidence when comparing results. We call for more studies directly comparing culture, GeneXpert and sputum smear microscopy to allow more accurate comparisons of diagnostic proof-of-concepts. No studies reported an approximate cost.

The lowest reported limit of detection for TB (≤10 cell mL^−1^) used Actiphage^®^ (PBD Biotech Ltd., Thurston, UK) [24], whereas the lowest (LOD50%: a 50% probability of detecting contamination at this level) for NTM was 10 cell 50 mL^−1^ by using phagomagnetic separation [25]. These two methods also had the lowest reporting times and highest sensitivities and specificities. Both methods targeted mycobacterial insertion sequence DNA with PCR, showing these to be good targets for accurate diagnostics.

Nucleic acid amplification tests were frequently used endpoints. Low limits of detection were achieved when mycobacterial insertion sequence DNA (IS6110) was targeted with PCR [24,25,26]. Given that multiple copies are present in a single cell, it is clear why they make a good target, especially for detecting small numbers of bacteria. Another method detected phage DNA [27] to good effect in drug susceptibility testing. 

Several assays used enzymes to catalyze the generation of their respective biomarkers. Using this method, two endpoints have been explored: the detection of changes in electric current [28,29] and the detection of unique nucleic acid sequences [30,31]. One approach used reporter phages to introduce the enzyme [31], whereas another used an enzyme already present in mycobacteria [30]. These methods had the highest limit of detection, but were still within the range of clinical relevance. The insensitivity of these methods may be due to their use of lytic phages, releasing cell contents and preventing further catalysis.

A reporter phage (Φ^2^GFP10) was developed to detect TB and rifampicin-resistance in LMICs [22]. During a trial in South Africa, the reporter phage could detect TB with a high degree of agreement in sensitivity and specificity compared to GeneXpert MTB/RIF in both smear-positive and smear-negative sputum samples. The ability to rapidly identify antimicrobial-resistant mycobacteria is also a great benefit, and by using this technology, extensively drug-resistant tuberculosis could be detected [32]. However, one drawback of this method was the need to carry out analysis using FACS, which reduces the ability to use this near to care in all high TB burden countries.

An emerging technology is the use of magnetic microbeads to capture bacilli followed by concentration using magnetic separation. This step does not require centrifugation or filtration and further inroads into automation. Historic efforts used peptide-beads followed by phage lysis [26,33]. The process has recently been streamlined by using phage-beads to capture and lyse in one step and using real-time PCR for the readout [25]. These methods had consistently low limits of detection. Inclusion of this step into other methods may improve accuracy, sensitivity and LMIC applicability. 

Few phage-diagnostics have been translated into commercial and clinical use. One way to ease this transition is to demonstrate the assay with clinical samples. Many proof-of-concept studies utilized clinical samples, improving confidence in their applicability [20,27,29,31,34]. A good example of proof-of-concept translation can be seen when the phage real-time PCR assay developed by Pholwat et al. [27] was implemented in a Thailand reference laboratory [35]. This allowed for direct comparisons against standard methods and demonstrated the assay’s capability in a high-volume, real-world setting. For the field of phage diagnostics to advance, more proof-of-concepts that are successful need to be developed through the translational pipeline.

Developing diagnostic tests for use in LMICs can be difficult, because tests need to be inexpensive to run and have access to a power source, largely limiting their use to reference laboratories. Isothermal amplification steps as well as the development of colorimetric assays have been explored to circumvent these issues and move towards a point-of-care test [30,36]. However, there have been difficulties experienced in achieving low limits of detection.

One inherent limitation of using phages as lysis agents comes from the time they take to lyse mycobacteria. For instance, D29 takes 3.5 h to enter the eclipse phage and burst target cells; a fundamental aspect of phage biology that is seemingly unavoidable. Delaying time to detection can be somewhat mitigated by using faster endpoint detection methods. For example, using colorimetric results as opposed to quantitative when only presence/absence information is needed.

The inconsistent burst size of phages can create limitations when detecting phage DNA in real time. Exact burst sizes vary; therefore, setting a threshold to differentiate between inoculated phage DNA and amplified phage DNA is tricky, resulting in difficulties in creating specific diagnostics for low levels of bacteria. Resolutions of this problem would advance the field of phage diagnostics.

Difficulties detecting low bacterial concentrations with phages arise from the low likelihood of phages randomly interacting with a single cell within a given space. Efforts to circumvent this include maximizing the multiplicity of infection [24]. Other methods have used magnetic beads [25,26,33], to capture and concentrate the bacteria, facilitating infection. These efforts have largely succeeded, seen in the low limits of detection reported. However, new diagnostics should be mindful of this pit fall.

Phage diagnostics can be improved by standardizing comparator tests and translating more successful proof-of-concepts. Working towards the WHO’s diagnostic guidelines [1] in the proof-of-concept stage will ease the transition. The field can improve by focusing efforts on developing point-of-care tests.

Bacteriophage-based diagnostics offer great potential. The advantages of phages are numerous; only viable bacilli are detected whereas specificity is determined by the phage’s host range. They can be produced at a low cost, are easy to handle, and their rapid rate of infection can drastically reduce reporting times. Due to their several advantages, phages may fulfil the needs of modern TB diagnostics.

Given that both phage-therapy and phage-diagnostics are becoming more prevalent, their interplay needs to be considered. We should always be mindful of resistance. In isolation, diagnostics circumvent this concern by operating as a closed system; however, resistance derived from therapy will likely impact diagnostics if the same phage is used in both instances. When designing and implementing these therapies and technologies, this needs to be considered.

## 3. Bacteriophages as a Treatment Option for Mycobacterial Infections

Antibiotic resistance is becoming a major public health issue throughout the world. The spread of multidrug-resistant (MDR) bacteria is a threat to human health and extensive antibiotic resistance has developed in various bacteria, including mycobacteria, due both to innate resistance in some species and the fact that some bacteria are highly adept at acquiring antibiotic-resistant determinants from each other or during treatment [37]. Bacteriophages have been proposed for decades for the treatment of common bacterial infections, but mycobacterial infections have generally been excluded [38]. We can only speculate that the availability of effective oral antimycobacterial drugs and the prolonged length of treatment for most of mycobacterial infections did not make intravenous mycobacteriophages an appealing field to be pursued. With the rise of drug resistance, their clinical utility has been re-discovered and their use has been successfully demonstrated, both as treatment options, but also as possible alternatives for the disinfection of water systems, in animal health and in the food industry [39,40,41,42]. Phage preparations have successfully been approved for use against *Escherichia coli* and *Listeria monocytogenes* in meat contamination [43]. Although phages have been extensively used therapeutically in former Soviet Union countries, their clinical use in the Western world is generally case-by-case, under the compassionate use route, when all other treatment options have failed. There have been no commercial preparations and there are no standard regulations on their use in the United States, leaving the choice of phage treatments as the last resort for very few desperate cases and relying on the ability and connections of clinical teams to source them. The first patient treated with intravenous phage therapy in the United States suffered from a systemic infection caused by a multidrug-resistant *Acinetobacter baumannii*, although a complete recovery was achieved [44]. Some case series on the compassionate use of phage therapy (PT) in Europe have also highlighted significant clinical improvement in the infections after phage treatment [45,46], and clinical trials on the use of PT to treat chronic otitis and other infections caused by *Pseudomonas aeruginosa*, *Staphylococcus aureus* and *Enterobacterales* have been published, with promising results [47,48,49]. However, the use of mycobacteriophages (phages active against different mycobacterial species) is still very limited, and no clinical trials have been reported to date.

*Mycobacterium* spp. belong to the family *Mycobacteriaceae* of the class *Actinobacteria*. The taxonomy of mycobacteria is regularly updated, and the most recent classification was released in 2017 with over 170 recognized species [50]. Based on phenotypic and genetic differences, the genus can be classified into two main groups: slowly growing mycobacteria, including *Mycobacterium tuberculosis* complex; and rapid growers, NTMs, that are generally environmental organisms that cause opportunistic infections [51]. The worldwide impact of TB has already been mentioned. Other NTM infections are also increasing worldwide, due to the expanding numbers of immunocompromised individuals (including those with HIV infection and hematological disorders), as well as patients with cystic fibrosis (CF) and chronic lung disorders [52,53]. Among the various NTMs, the MAB complex comprises a group of rapidly growing, multidrug-resistant mycobacteria that are responsible for a wide spectrum of skin and soft tissue diseases, lung and central nervous system infections, bacteremia, ocular and other infections. These infections are often problematic and difficult to treat, due to the innate resistance of MAB to many antibiotics [54,55]. Typically, patients are treated with last-line antibiotics that have extreme toxicities for an extended amount of time (~12 months) and, in many cases, they cannot tolerate the side-effects of these drug regimens.

Phage activity has been successfully demonstrated against different mycobacterial species, including *M. abscessus* [56], *M. ulcerans* [57], *M. avium* and *M. tuberculosis* [58,59,60]. However, most of the data is based on laboratory and animal models and reports of clinical cases are very scant thus far. The first clinical case of phage treatment (PT) against drug-resistant MAB was a 15-year-old patient suffering from cystic fibrosis, who had a double lung transplant and persistent disseminated infection [61]. The treatment consisted of a cocktail of three different phages administered intravenously (IV), two of which were temperate and genetically engineered to convert into lytic phages, twice daily, at 10^9^ PFU/dose [61]. Most importantly, the administration of the phage IV was safe, with no toxicities or side-effects. After 6 weeks of treatment, complete resolution of an infected liver node was seen, along with an increase in the patient’s weight and lung function. Skin nodules were slower to improve. After 121 days of treatment, the patient’s MAB isolates were still susceptible to each of the three phages in the cocktail, and a neutralizing immune response was not seen. In a more recent case study, using the same three-phage cocktail, a phage-neutralizing antibody response was demonstrated to be the cause of phage treatment failure [62]. The patient, an immunocompetent 81-year-old male with MAB lung disease, was given the cocktail IV twice daily at 10^9^ PFU/dose. Colony-forming unit (CFU) counts from sputum revealed a 1-log decrease in MAB after one month of PT. However, MAB CFU then increased steadily from two to six months of PT. The patient’s MAB samples were still fully susceptible to two of the phages used in the cocktail, but had an intermittent 1–2 log decrease in susceptibility to one phage. Patient serum revealed a strong neutralizing antibody response to all three phages in the cocktail, which was further analyzed using ELISAs and found to be primarily IgG-mediated [62]. 

PT for patients with MAB infections is complicated, because strain variability is extensive [62] and phage susceptibility is unpredictable due to MAB variations. Dedrick et al. [62] found that colony morphotypes (smooth or rough) influence phage susceptibility, because smooth isolates are more resistant to phage infection. Additionally, clinical MAB isolates have various prophages (1–6 per strain) integrated into their genomes, which can influence phage susceptibility due to phage defense systems [63,64].

A recent review of all the clinical requests for PT at the Center for Innovative Phage Applications and Therapeutics (IPATH) in San Diego showed that over a two-year period, there were 90 requests for PT against different mycobacterial infections (47 *M. abscessus*, 23 *M. avium*, 7 *M. chimaera*, and 13 other Mycobacterium species, including *M. chelonae, M. smegmatis, M. xenopi*, and *M. genavense*). However, PT was approved and administered to only four patients with *M. abscessus* infection (with a further three patients pending administration at the time of publication) and only one patient with *M. chimaera* [65]. In all cases, PT was given intravenously with topical administration in patients with skin lesions. 

Antibiotic resistance in *M. tuberculosis* is certainly a growing concern, particularly with the emergence of extensively drug-resistant (XDR) and totally drug-resistant (TDR) strains. Successful XDR-TB treatment, particularly in resource-limited settings, may be very challenging. In a 2006 XDR-TB outbreak in KwaZulu-Natal, South Africa, 52 of 53 people who contracted the disease died within months [66], even with survival rates significantly improving in more recent years [67]. Delivery of phages to the lungs could benefit from aerosolization; however, it is uncertain whether phages could target intracellular or intra-granuloma *M. tuberculosis* as well as extracellular *M. tuberculosis* [68]. Although phages may not be taken up directly by macrophages, they may be dynamically cycled among the broader population by piggybacking on the natural bacterium–macrophage dynamics [69]. The use of a nonvirulent mycobacterium, specifically, *M. smegmatis*, has been proposed as both a potential delivery system (carrier) and as a bacterial host that can lead to the high proliferation of bacteriophages [60]. Recently, some authors have demonstrated that a cocktail of three to five different mycobacteriophages was effective against *M. smegmatis* under low-pH, hypoxic and stationary conditions (mimicking the granuloma) and showed synergy with rifampicin. Similarly, they have found that three mycobacteriophages (DS6A, D29 and TM4) were also able to prevent the growth of *M. tuberculosis* [59]. The same D29 mycobacteriophage (already mentioned for diagnostic testing and treatment option) has also been proposed as potential prophylaxis to prevent tuberculosis in the mice model with an optimized inhalation device. The bacteriophage aerosol pre-treatment significantly decreased the *M. tuberculosis* burden in mouse lungs at 24 h and 3 weeks post-challenge [70,71].

In addition to the phages already mentioned, other bacteriophages have also been investigated against *M. tuberculosis* as therapeutic options (including phages TM4, T7, P4, PDRPv, BTCU-1, Bo4, SWU1, GR-21/T, My-327, Ms6 and Bxz2) [72]. More recently, colleagues from Pittsburgh have assembled a five-phage cocktail that minimized the emergence of phage resistance and cross-resistance to multiple phages (AdephagiaΔ41Δ43, D29, FionnbharthΔ45Δ47, Fred313_cpmΔ33, and Muddy_HRMN0157-2), and which efficiently killed a series of *M. tuberculosis* reference strains representing its common lineages [58].

Other authors have also been successful in encapsulating mycobacteriophages into giant liposomes and showing their uptake into eukaryotic cells more efficiently than free bacteriophages [73]. This could represent an ideal formulation for inhaled administration, as recently demonstrated for liposomal amikacin in the treatment of NTM lung infection [74].

The administration of treatment, the need for multiple active phages and the potential development of resistance are only some of the multiple challenges that phage therapy against mycobacterial infections still needs to overcome [42] (Table 2). More than 18,000 actinobacteriophages have been described; the selection of an active phage is a laborious process [75]. Due to phage specificity, tailored treatments are necessary and very few centers in the world are able to perform this personalized manufacturing, causing significant delays from request to administration. Resident prophages may strongly influence the phage infection profiles and influence which phages are therapeutically useful [63], and some bacterial strains, such as the smooth morphotypes of *M. abscessus*, may be intrinsically resistant [56]. The production of neutralizing antibodies to the phages can also significantly limit their therapeutic efficacy. All bacteriophages are capable of inducing a specific antibody response (IgM and IgG), as demonstrated in animal models and as observed in humans. These antibodies might impact phage bioavailability, although further in vivo studies are needed to assess the impact on treatment outcomes [76]. Hence, the selection process of active phages is far from an easy task.

It is important to note that the concept of phage resistance is different from the mechanisms of resistance against antibiotics. For example, the phage resistance in vitro is very pathogen and phage-specific, and it is not widely transferable, such as the extended-spectrum-β-lactamase resistance observed in Gram-negative bacteria [77]. For MAB, it has been observed at a very low level [56] and it may lead to different degrees of resistance in vivo relative to in vitro. It can be hypothesized that phage-resistant mutants of *M. tuberculosis* might be less fit due to a loss of virulence. This will inevitably influence therapeutic strategies, where phage monotherapy may be plausible without resistance being a major concern in MAB, but the impact of PT on *M. tuberculosis* still needs to be assessed in clinical practice. If the development of resistance in TB is not going to be a relevant problem, it raises the possibility of using phages for long-term treatments, where the goal is to suppress active disease and dissemination rather than to effect a ‘cure’ as the outcome in MDR compassionate cases or where nebulized bacteriophages can be used as an adjuvant strategy in addition to antibiotics with the aim of shortening the overall length of treatment to only few months.

It is also important not to forget the necessary regulatory process. Phage therapy remains experimental; therefore, each clinical case will require multiple local approvals (i.e., ethical committee, FDA in the United States, NHSE in the United Kingdom and other national bodies in different European countries). The majority of compassionate cases generally include evidence of clinical need and failure of previous treatments, proof of in vitro bacterial susceptibility to the phage(s), sequencing and genetic characterization of the phage(s), with particular focus on delineating any potential risk of transmitting plasmids and genetic material encoding for resistance mechanisms (both in phages and bacteria), lack of lysogenic activity, sterility of the final product and minimal endotoxin concentration [65,78,79]. Recently, colleagues from San Diego have proposed a standardized bacteriophage purification protocol for personalized phage therapy (requiring 16–21 days in total), with a systematic procedure for phage isolation, liter-scale cultivation, concentration and purification [80]. Despite the various challenges, mycobacteriophages are a promising alternative option for the treatment of mycobacterial infections, and further research and future clinical trials are needed to assess their role as adjuvants in order to reduce the total duration of treatment.

## 4. Conclusions

This narrative review has provided a broad overview on the current use of bacteriophages for both the diagnosis and treatment of mycobacterial infections. Different bacteriophage-based diagnostics have been developed which offer great potential, being rapid, low-cost and easy to handle. A couple of commercial assays are already available and various proof-of-concept technologies have recently been published, hopefully moving soon to the next stage of development. If successfully implemented, phages may fulfil the needs of modern TB and NTM diagnostics, allowing rapid detection, control and early treatment.

Mycobacteriophages have also been used for the treatment of challenging infections, mostly multidrug-resistant *M. abscessus*. The few case reports and small series have demonstrated some promising outcomes and, most importantly, the safety of phage treatment. However, the process is far from being straightforward, from the selection of active phages to regulatory approval, and it is still only limited to compassionate cases. Extensive clinical trials are needed to assess their clinical advantages. Mycobacteriophages against *M. tuberculosis* have been selected, but they are still in the discovery/early pre-clinical phase.

Finally, a note of caution to conclude. Drug resistance in *M. tuberculosis* developed shortly after the introduction of streptomycin, para-aminosalicylic acid and isoniazid as first line treatments. Given that both phage-therapy and phage-diagnostics are becoming more prevalent, their concomitant use needs to carefully consider the risk of resistance and how to address it to preserve their utility for mycobacterial infections. More studies are also needed to better understand the development of neutralizing antibodies, and their relation to the phage structure or any predisposing human genes. The mycobacteriophage journey in the Western world has just started.

## Figures and Tables

**Table 1 microorganisms-09-02366-t001:** Phage technologies used to detect mycobacterial infections. * MTB, *M. tuberculosis*; MAP, *M. avium* subspecies paratuberculosis.

Commercial Assays Already Available								
Name	Mechanism of Action	*Mycobacterium* spp.	Phage(s) Used	Limit of Detection	Sensitivity	Specificity	Turnaround Time	References
Actiphage^®^ Rapid (PBD Biotech Ltd., Thurston, UK)	Mycobacteria are isolated from peripheral blood mononuclear cells, then the phage is used as a lysis agent. PCR, detecting mycobacteria, is used as an endpoint.	MTB *, MAP *, *M. bovis*	D29	≤10 cell mL^−1^	95%	100%	6 h	[24]
FASTPlaque TB™ (Biotech Labs Ltd., Ipswich, UK)	Phage amplified biologically assay. Other kits (FASTPlaque RIF™/MDR™) offer drug susceptibility testing.	MTB	D29	100–300 cell mL^−1^	95%	95%	48 h	[16,21]
**Proof-of-concept Assays**								
Enzyme detection biosensor	Phages are used as a lysis agent. The released enzyme (TOP1A) binds and cleaves a surface bound DNA complex. Addition of Mg^2+^ causes DNA circularization and enzyme turnover. The DNA circle is amplified by rolling circle amplification. Then, visualized using fluorescent probes.	*M. smegmatis*	D29; Adephagia Δ41, Δ43	0.6 million CFU mL^−1^	-	100%	-	[30]
Phage real-time PCR	48 h pre-incubation with first- and second-line antibiotics. Then, incubated with phage. Real-time PCR used to detect phage DNA. Extracellular phages are inactivated. Presence of phage indicates cell viability, and thus, resistance. Later adapted so that real-time PCR is directly performed on MGIT broths for clinical applicability.	MTB	D29	-	90%	99%	1 to 3 days (proof-of-concept)/positive MGIT culture plus 3 days (clinical)	[27,35]
Phagomagnetic separation	Phage-coated paramagnetic beads capture and concentrate bacilli. Bead-bound mycobacteria are separated using magnetism. Mycobacterial DNA is released (phage-mediated lysis) and detected by real-time PCR.	MAP	D29	LOD_50%_: 10 cell 50 mL^−1^	97%	99%	7 h	[25]
Peptide mediated magnetic separation	Bead-bound peptides capture and concentrate bacilli, which are then separated magnetically. Then, the phage-amplified biologically assay, followed by plaque PCR, are used for detection.	MAP	D29	10 cell mL^−1^	-	-	48 h	[26]
Electrochemical detection of enzymatic action	Phages are used as a lysis agent. The activity of a released enzyme (beta-glucosidase) is quantified amperometrically.	*M. smegmatis*	D29	10 cell mL^−1^	-	-	8 h	[28]
Surrogate marker locus generation module	16 h pre-incubation with first- and second-line antibiotics. Phage encoded with RNA cyclase ribozyme, under SP6Pol transcriptional control, generate circular surrogate marker locus RNA. This unique nucleic acid sequence is detected by reverse transcriptase PCR. Presence of surrogate marker locus RNA indicates cell metabolic activity, and thus, resistance.	MTB	phSGM2	<100 CFU	-	-	1 to 2 days	[31]
Peptide-mediated magnetic separation with phage ELISA	Bead-bound peptides capture and concentrate bacilli, which are then separated magnetically. This concentrate is incubated with phage. Extracellular phages are inactivated. D29-specific ELISA is used as an endpoint.	MAP	D29	~100 PFU mL^−1^	-	-	<1 day	[33]
Phage-amplified multichannel series piezoelectric quartz crystalsensor	Phage-amplified biologically assay performed in liquid broth. The response curve of the reporter *M. smegmatis* is measured using a multichannel series piezoelectric quarts crystal sensor.	MTB	D29	100 CFU mL^−1^	89%	95%	30 h	[29]
Colorimetric detection testing phage replication	Mycobacteria are added to a 96-microwell plate with antibiotics and incubated overnight. Phage is added. After incubation, extracellular phage are inactivated. Samples were added to a fresh 96-microwell plate containing reporter *M. smegmatis* and incubated overnight. A redox dye, MTT, was added. Growth of *M. smegmatis* results in a color change. Lack of a color change indicates lysis of *M. smegmatis* by phage, and thus, the viability of mycobacteria in the original 96-microwell plate.	MTB	D29	-	91%	99%	>2.5 days	[36]
Fluorescent Reporter Phage	GFP-modified mycobacteriophage are incubated with a processed sputum sample and fluorescence indicates the presence of a viable mycobacterial host. Fluorescence is detected by FACS	MTB	Φ^2^GFP10	<10^4^	96%	83%	>2 days	[22]

* MTB = Mycobacterium tuberculosis − MAP = Mycobacterium avium subsp. Paratuberculosis; (-) = No data available.

**Table 2 microorganisms-09-02366-t002:** Challenges of phage therapy against mycobacterial infections, from the selection of phages, the necessary regulatory approvals and treatment considerations.

Challenges of Phage Therapy Against Mycobacterial Infections	
Selection of phages	Laborious screening process of thousands of different phages A cocktail of 3 to 6 active phages may be needed Few centers in the world are able to perform this personalized manufacturing
Administration	Intravenous route for disseminated infection is required Topical administration for skin lesion is easily performed *Still under development*: Use of a nonvirulent mycobacterium as carrier to reach the lung Liposomal formulations for inhalation in case of lung infection
Development of resistance	Intrinsically resistance strain (i.e., smooth morphotype of *M. abscessus*) Acquired resistance after treatment/bacterial defense mechanisms Production of neutralizing antibody against the phages
Regulatory process	Each clinical case required multiple local approvals, including ethical committee and national approval body Genetic characterization of the phage(s), sterility of the final product and minimal endotoxin concentration required prior to approval

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
