# Peer review of "Application of Bacteriophages for Mycobacterial Infections, from Diagnosis to Treatment"

_microorganisms, 2021, doi:10.3390/microorganisms9112366_

Round 1

Reviewer 1 Report

Shield et al. offers a synopsis of the recent developments in the use of bacteriophages as a diagnostic tool and as a therapeutic intervention for Mycobacterial infections. The authors first review how mycobacterial phages have been utilized for the detection and diagnosis of Mycobacteria spp., emphasizing the need for products that can be utilized in the setting of low- to middle-income countries, which have a high burden of infections. The second part of the review elaborates on clinical outcomes of bacteriophage therapy, including recent successes, failures, and current challenges. Overall, the manuscript is well-organized and clearly written.

One area that I found superficial was the description of bacteriophage pharmacology in general. How must phage be prepared to prevent toxicity (mentioned on line 222)? What else is known about the interaction of phages with the immune system, besides the Dedrick et al. 2021 study? Are mycobacteriophages only effective against mycobacteria outside human cells?

Some other comments:

Lines 42 -43 : MSMDs acronym isn’t defined. BGC acronym is used before defining it.

Line 52-53: reference needed.

Line 62-64: reference needed.

Line 78-79: awkward sentence.

Line 90-91: missing the word “cells” after “break open”.

Line 112-113: define the acronym “LOD50%”.

Line 162: misspelling of “numerous”.

Line 272: “Different other bacteriophages” awkward sentence.

Line 316: “genetic characterization of the phage(s) with particular focus on plasmids encoding for resistance mechanisms,” unclear as stated. Are the regulators interested in identifying genes horizontally transferred via plasmids that confer resistance to phages?

Reviewer 2 Report

Dear Editor,

Shield et al have presented a comprehensive review on the use of bacteriophages for the diagnosis and treatment of Mycobacterial infections. In this review, the authors briefly summarize the current diagnostic options using bacteriophage including proof of concept assays. Next, the authors elaborate on how phage therapy is not so extensively used in the western world and provide successful pieces of evidence of phage therapy against various Mycobacterial species. Finally, the authors conclude with the challenges of phage therapy and what measures need to be considered before using phage therapy as the first line of treatment.  

The authors have performed a comprehensive review of the literature and presented it with clarity. I do have some minor suggestions that the authors could briefly elaborate on. These are outlined below.

Line 62-64: The authors state how the GeneXpert system has made a difference in TB diagnosis. It would be useful to briefly highlight how the GeneXpert system helps with TB diagnosis

Line 90-93: The authors mention both PhAB and PRA assays as the commercially used assays for TB diagnosis. It would useful to describe how the assay works, which phage is used before stating its limitation.

Line 108-109: Could the authors clarify which comparators were used?

Line 113-114: It would be useful to state the two methods along with the citation for better clarity.

Line 130: Please indicate the reporter phage ɸ2GFP10

Line 163: It would be helpful to outline some of the limitations of phage-based diagnostics and provide the author’s perspective on how phage-based diagnostics can be improved and used as a standardized technology

Table 1: The authors could also briefly mention luciferase-based reporter assays that were used (Dusthackeer: J Microbiol Methods 73 (1), 18- 25 (2008) ) which was reported to be more accurate than Fastplaque assays.

Line 180: Could you elaborate on why mycobacterial infections have been excluded for phage-based therapy.

Line 187-188: The authors could elaborate on why phage therapy is used only as a last resort for treatment for example not knowing the exact dosage for treatment with phage therapy, the need to identify the exact phage to treat infection, some phages make bacteria more resistant. The authors have explained this in lines 314-318. It would be useful to include it here instead.

Line 250: Could the authors state the outcome of the PT in patients with skin infections and how it was beneficial.

Lines 268-271: It is interesting to emphasize how the same D29 phage is used for diagnostic testing, treatment, and prophylaxis.
